# *couple*CoC+: An information-theoretic co-clustering-based transfer learning framework for the integrative analysis of single-cell genomic data

Pengcheng Zeng◉, Zhixiang Lin◉*

Department of Statistics, The Chinese University of Hong Kong, Hong Kong SAR, China

* zhixianglin@cuhk.edu.hk

## Abstract

Technological advances have enabled us to profile multiple molecular layers at unprecedented single-cell resolution and the available datasets from multiple samples or domains are growing. These datasets, including scRNA-seq data, scATAC-seq data and sc-methylation data, usually have different powers in identifying the unknown cell types through clustering. So, methods that integrate multiple datasets can potentially lead to a better clustering performance. Here we propose *couple*CoC+ for the integrative analysis of single-cell genomic data. *couple*CoC+ is a transfer learning method based on the information-theoretic co-clustering framework. In *couple*CoC+, we utilize the information in one dataset, the source data, to facilitate the analysis of another dataset, the target data. *couple*CoC+ uses the linked features in the two datasets for effective knowledge transfer, and it also uses the information of the features in the target data that are unlinked with the source data. In addition, *couple*CoC+ matches similar cell types across the source data and the target data. By applying *couple*CoC+ to the integrative clustering of mouse cortex scATAC-seq data and scRNA-seq data, mouse and human scRNA-seq data, mouse cortex sc-methylation and scRNA-seq data, and human blood dendritic cells scRNA-seq data from two batches, we demonstrate that *couple*CoC+ improves the overall clustering performance and matches the cell subpopulations across multimodal single-cell genomic datasets. *couple*CoC+ has fast convergence and it is computationally efficient. The software is available at https://github.com/cuhklinlab/coupleCoC_plus.

## Author summary

The recent advances in single-cell technologies have enabled multiple biological layers to be probed and provides unprecedented opportunities to assay cellular heterogeneity. To analyze the complex biological processes varying across cells, we need to obtain and integrate different types of genomic features through flexible but rigorous computational methods. The most important challenge for data integration is to link data from different sources in a way that is biologically meaningful. In this work, we have developed a transfer

**Data Availability Statement:** All data and source code are publicly available. The mouse cortex scRNA-seq data in example 1 and example 3 are available at NCBI Gene Expression Omnibus (GEO)

under accession GSE115746. The mouse cortex scATAC-seq data in example 1 were downloaded from https://atlas.gs.washington.edu/mouse-atac/data/. The mouse and human scRNA-seq data in example 2 are available at https://panglaodb.se/view_data.php?sra=SRA832392&srs=SRS4237518 and https://panglaodb.se/view_data.php?sra=SRA878024&srs=SRS4660846, respectively. The mouse cortex sc-methylation data in example 3 are available at GEO under accession GSE97179. The human blood dendritic cells scRNA-seq data in example 4 are available at GEO under accession GSE94820. Source code are available at https://github.com/cuhklinlab/coupleCoC_plus.

**Funding:** Both PZ and ZL are supported by the Chinese University of Hong Kong direct grants No. 4053360 and No. 4053423, the Chinese University of Hong Kong startup grant No. 4930181, and Hong Kong Research Grant Council Grant ECS No. CUHK 24301419, and GRF No. CUHK 14301120. The funders had no role in study design, data collection and analysis, decision to publish, or preparation of the manuscript.

**Competing interests:** The authors have declared that no competing interests exist.

learning method based on the information-theoretic co-clustering framework for the integrative analysis of single-cell genomic data. This method utilizes the information from one dataset to boost the analysis of another dataset, and it also uses the information of the features that are unlinked in the two datasets. We demonstrate that our transfer learning-based clustering method significantly improves clustering performance in single-cell genomic datasets. Our results show that transfer learning is promising for the integrative analysis of single-cell genomic data.

This is a *PLOS Computational Biology* Methods paper.

## Introduction

The advances in single-cell technologies have enabled the profiling of multiple molecular layers and have provided great opportunities to study cellular heterogeneity. These technologies include single-cell RNA sequencing (scRNA-seq) that profiles transcription, single-cell ATAC sequencing (scATAC-seq) that profiles accessible chromatin regions [1–3], single-cell methylation assays that profile methylated regions [4–7] and other methods. The datasets [8–10] brought by these technologies lead to increasing demands for computationally efficient methods for processing and analyzing the data. However, single-cell genomics data often have high technical variation and high noise level due to the minimal amount of genomic materials isolated from individual cells [11–14]. These experimental factors bring the challenge of analyzing single-cell genomic data, and affect the results and interpretation of unsupervised learning methods, including dimension reduction and clustering [15–18].

Clustering methods, which group similar cells into sub-populations, are often used as the first step in the analysis of single-cell genomic data. Most clustering methods are designed for clustering one type of measurement. The clustering methods for scRNA-seq data include SIMLR [19], SC3 [20], DIMM-SC [21], SAFE-clustering [22], SOUP [23], SAME-clustering [24] and SHARP [25]. The methods *chrom*VAR [26], *sc*ABC [27], SCALE [28], *cis*Topic [29] and Cusanovich2018 [30] are developed for analyzing scATAC-seq data. Clustering methods have also been proposed for methylation data [31, 32]. To comprehensively analyze the complex biological processes, we need to acquire and integrate different types of measurement from multiple experiments. In recent years, some methods are developed for this purpose. They include Seurat [33, 34], MOFA [35], *couple*NMF [36], scVDMC [37], LIGER [38], *sc*ACE [39], MOFA+ [40], scAI [41], *couple*CoC [42] and scMC [43]. A more comprehensive discussion on integration of single-cell genomic data is presented in [44].

To link data from different sources in a way that is biologically meaningful is the most important challenge in the integration of single-cell data across different types of measurement. As an example, we consider the setting where scRNA-seq and scATAC-seq are profiled on similar cell subpopulations but different cells. It is desirable to utilize the information in scRNA-seq data to help us cluster scATAC-seq data, which is typically sparser and noisier. A subset of features in scATAC-seq data are linked with scRNA-seq data, because promoter accessibility/gene activity score are directly linked with gene expression. The linked features help us connect the two data types, which may lead to improvement in clustering scATAC-seq data. Besides the linked features, we can also leverage the unlinked features in the scATAC-seq data: accessibility of the peaks distant from the genes is not directly

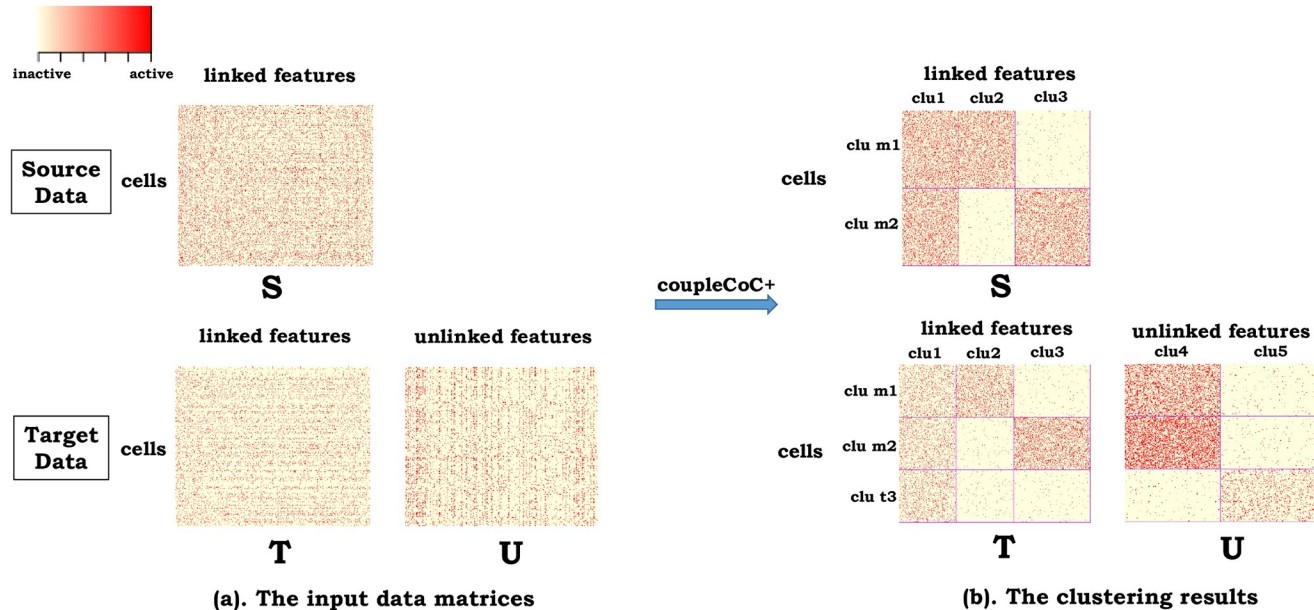

**Fig 1. Toy example of *couple*CoC+.** (a). Source data is represented by "S". Based on whether the features are linked with those in the source data, we partition the target data into two parts, "T" and "U". The features in data T are linked with data S, while the features in data U are not directly linked with data S. The cells in data T and U are the same. Red color means that the corresponding features are active, and yellow color means that they are inactive. (b). The clustering results by *couple*CoC+. *couple*CoC+ co-clusters the data S, T and U simultaneously by clustering similar cells and similar features. A subset of the cell clusters are also matched between the source data and the target data, representing shared cell types. "clu" is the abbreviation of "cluster", and "m" means the matched clusters. "clu t" represents the cell cluster that is unique to the the target data.

linked with gene expression in scRNA-seq data. Incorporating more information by including the unlinked features is expected to further improve the clustering performance of the scATAC-seq data.

In this work, we propose *couple*CoC+, which is based on the information-theoretic co-clustering [45] transfer learning framework for the integrative analysis of single-cell genomic data (Fig 1). The goal of *couple*CoC+ is to utilize one dataset, the source data (S), to facilitate the analysis of another dataset, the target data. Depending on whether the features are linked with the source data or not, the target data can be partitioned into two parts, data T with the linked features, and data U with the unlinked features (Fig 1(a)). As an example, we may use scRNA-seq data as the source data S and scATAC-seq data as the target data. Data T is the data matrix of gene activity score, which are directly linked with gene expression in scRNA-seq data, and data U is the data matrix for the accessibility of peaks distal to the genes, which are not directly linked with gene expression. *couple*CoC+ not only transfers information from the source data, but also utilizes information from the unlinked features in data U. In *couple*CoC+, both the genomic features and the cells are clustered (Fig 1(b)). The key for knowledge transfer between the source data and the target data is that the cluster assignments for the linked features are the same. *couple*CoC+ also performs matching of a subset of cell clusters across the source data and the target data, which may represent shared cell types across the two datasets. We refer our model as *couple*CoC+, because it is based on the framework of our previously proposed *couple*CoC [42]. *couple*CoC+ addresses the limitations of *couple*CoC by including the unlinked features in target data and it also integrates co-clustering and cell type matching in one step for better use of information from the source data.

## Materials and methods

In this section, we first introduce the information-theoretic co-clustering framework for source data [45], and then extend it to our framework of co-clustering source data and target data simultaneously. We will choose the less noisy dataset as the source data, such as scRNA-seq data, and we will choose the noisier dataset as the target data, such as scATAC-seq data and sc-methylation data. We assume that a subset of features in the target data are linked with the source data: gene activity score in scATAC-seq data and gene body methylation in sc-methylation data are linked with gene expression in scRNA-seq data; and the other subset of features are not directly linked: peak accessibility in scATAC-seq data and DNA methylation levels at non-CG sites for non-overlapping bins in sc-methylation data are not directly linked with the genes in scRNA-seq data. Promoter accessibility may also be used to link scATAC-seq data with scRNA-seq when gene activity score is not available. Promoter accessibility may have less power in separating the cell types compared with gene activity score, because gene activity score incorporates more regions nearby the gene. We expect to improve the clustering performance of the target data by transferring knowledge from the source data via the linked features and also effectively utilizing the information in the unlinked features in the target data.

### Information-theoretic co-clustering

We first consider the source data. Let S be a $n_S$ by $q$ matrix representing this dataset with $q$ features for $n_S$ cells. Let $X$ and $Z_S$ be discrete random variables, representing the possible outcome of cell labels and feature labels, respectively. $X$ takes values from the set $\{1, 2, \ldots, n_S\}$ and $Z_S$ takes values from the set $\{1, 2, \ldots, q\}$. We let $p_S(X, Z_S)$ be the joint probability distribution for $X$ and $Z_S$, and define $p_S(X = x, Z_S = z)$ as the probability of the $z$-th feature being active in the $x$-th cell: the more active the feature, the higher the value. This joint probability is estimated from the normalized dataset, i.e. scaling the data matrix S to have total sums equal to 1, and we have $p_S(X = x, Z_S = z) = \frac{S_{xz}}{\sum_{x=1}^{n_S} \sum_{z=1}^{q} S_{xz}}$, where $x \in \{1, \ldots, n_S\}, z \in \{1, \ldots, q\}$.

The goal of co-clustering is to cluster similar cells into clusters and similar features into clusters. Assume that we want to cluster the cells into $N_S$ clusters, and the features into $K$ clusters. We denote the clusters of cells and features as the possible outcomes of discrete random variables $\tilde{X}$ and $\tilde{Z}_S$, where $\tilde{X}$ and $\tilde{Z}_S$ take values from the sets of cell cluster indexes $\{1, \ldots, N_S\}$ and feature cluster indexes $\{1, \ldots, K\}$, respectively. To map cells to cell clusters and features to feature clusters, we use $C_X(\cdot)$ and $C_Z(\cdot)$ to represent the clustering functions of cells and features, respectively. $C_X(x) = \tilde{x}$ ($\tilde{x} = 1, \ldots, N_S$) indicates that cell $x$ belongs to cluster $\tilde{x}$, and $C_Z(z) = \tilde{z}$ ($\tilde{z} = 1, \ldots, K$) indicates that feature $z$ belongs to cluster $\tilde{z}$. We then let $\tilde{p}_S(\tilde{X}, \tilde{Z}_S)$ be the joint probability distribution for $\tilde{X}$ and $\tilde{Z}_S$, and this distribution can be expressed as

$$\tilde{p}_S(\tilde{X} = \tilde{x}, \tilde{Z}_S = \tilde{z}) = \sum_{x \in \{C_X(x) = \tilde{x}\}} \sum_{z \in \{C_Z(z) = \tilde{z}\}} p_S(X = x, Z_S = z). \tag{1}$$

Note that $\tilde{p}_S(\tilde{X} = \tilde{x}, \tilde{Z}_S = \tilde{z})$ is connected to $p_S(X, Z_S)$ via the clustering functions $C_X(\cdot)$ and $C_Z(\cdot)$. The matrix $\tilde{p}_S(\tilde{X}, \tilde{Z}_S)_{N_S \times K}$ can be interpreted as the low dimension representations for the cell clusters in the source data S.

The information-theoretic co-clustering [45] aims at finding the optimal clustering functions $C_X(\cdot)$ and $C_Z(\cdot)$ to minimize the loss of mutual information:

$$\ell_S(C_X, C_Z) = I(X; Z_S) - I(\tilde{X}; \tilde{Z}_S), \tag{2}$$

where $I(\cdot)$ denotes the function of mutual information, and we have
$I(X; Z_S) = \sum_x \sum_z p_S(x,z) \log \frac{p_S(x,z)}{p_S(x)p_S(z)}$, and $I(\tilde{X}; \tilde{Z}_S) = \sum_{\tilde{x}} \sum_{\tilde{z}} \tilde{p}_S(\tilde{x}, \tilde{z}) \log \frac{\tilde{p}_S(\tilde{x},\tilde{z})}{\tilde{p}_S(\tilde{x})\tilde{p}_S(\tilde{z})}$.

## The framework of *couple*CoC+

We now extend the information-theoretic co-clustering framework to multiple datasets, and simultaneously perform matching of the cell types across datasets (see the toy example in Fig 1).

Besides the source data S, we have another target data. The goal of *couple*CoC+ is to improve the clustering performance of the target data, utilizing the information in the source data. Depending on whether the features are linked with the source data, the target data can be partitioned into two parts: data T, which includes the linked features; and data U, which includes the unlinked features. Similar to the corresponding definitions for source data, we have the loss of mutual information for co-clustering the data T:

$$\ell_T(C_Y, C_Z) = I(Y_T; Z_T) - I(\tilde{Y}_T; \tilde{Z}_T), \tag{3}$$

where $Y_T$ and $Z_T$ are the discrete random variables representing the cell labels and the feature labels in data T, respectively. We have $I(Y_T; Z_T) = \sum_y \sum_z p_T(y,z) \log \frac{p_T(y,z)}{p_T(y)p_T(z)}$. $\tilde{Y}_T$ and $\tilde{Z}_T$ are the discrete random variables representing the cell cluster labels and the feature cluster labels in data T, respectively. We have $I(\tilde{Y}_T; \tilde{Z}_T) = \sum_{\tilde{y}} \sum_{\tilde{z}} \tilde{p}_T(\tilde{y}, \tilde{z}) \log \frac{\tilde{p}_T(\tilde{y},\tilde{z})}{\tilde{p}_T(\tilde{y})\tilde{p}_T(\tilde{z})}$. $C_Y$ and $C_Z$ are the clustering functions for the cells and the features in data T, respectively. Note that we assume that the feature clustering function $C_Z$ is the same for the linked features in data S and T. The function $C_Z$ is the key for knowledge transfer between source data and target data. By clustering similar features using the information from the source data, it effectively reduces the noise in the target data.

We also have the loss of mutual information for co-clustering the data U:

$$\ell_U(C_Y, C_U) = I(Y_U; Z_U) - I(\tilde{Y}_U; \tilde{Z}_U), \tag{4}$$

where $Y_U$ and $Z_U$ are the discrete random variables representing the cell labels and the feature labels in data U, respectively. We have $I(Y_U; Z_U) = \sum_y \sum_u p_U(y,u) \log \frac{p_U(y,u)}{p_U(y)p_U(u)}$. $\tilde{Y}_U$ and $\tilde{Z}_U$ are the discrete random variables representing the cell cluster labels and the feature cluster labels in data U, respectively. We have $I(\tilde{Y}_U; \tilde{Z}_U) = \sum_{\tilde{y}} \sum_{\tilde{u}} \tilde{p}_U(\tilde{y}, \tilde{u}) \log \frac{\tilde{p}_U(\tilde{y},\tilde{u})}{\tilde{p}_U(\tilde{y})\tilde{p}_U(\tilde{u})}$. $C_Y$ and $C_U$ are the clustering functions for the cells and the features in data U, respectively. Because the cells in data U and T are the same, data U and T share the same cell clustering function $C_Y$.

The matrices $\tilde{p}_S(\tilde{X}, \tilde{Z}_S)$ and $\tilde{p}_T(\tilde{Y}_T, \tilde{Z}_T)$ can be interpreted as the low dimension representations for the cell clusters in the source data and the target data. A subset of the clusters in the source data and target data may be matched, representing similar cell types across the two datasets. We denote $h_{T,N_{sub}}$ as a permutation of size $N_{sub}$ for the indexes of the cell clusters in data T, and denote $h_{S,N_{sub}}$ as an ordered permutation of size $N_{sub}$ for the indexes of the cell clusters in source data S. We then use $D_{KL}(\hat{p}_T(\tilde{Y}_{h_{T,N_{sub}}}, \tilde{Z}_T) \| \hat{p}_S(\tilde{X}_{h_{S,N_{sub}}}, \tilde{Z}_S))$ to measure the statistical distance between two probability distributions $\hat{p}_T(\tilde{Y}_{h_{T,N_{sub}}}, \tilde{Z}_T)$ and $\hat{p}_S(\tilde{X}_{h_{S,N_{sub}}}, \tilde{Z}_S)$, where $D_{KL}(\cdot \| \cdot)$ is Kullback-Leibler (KL) divergence [46]. These two distributions are obtained by extracting the rows $h_{T,N_{sub}}$ and $h_{S,N_{sub}}$ from $\tilde{p}_T(\tilde{Y}_T, \tilde{Z}_T)$ and $\tilde{p}_S(\tilde{X}, \tilde{Z}_S)$ correspondingly and then scaling the two submatrices to have summation equal to 1. The smaller the KL divergence, the more similar the subsets of cell clusters chosen by $h_{T,N_{sub}}$ and $h_{S,N_{sub}}$.

To co-cluster the three data T, S and U simultaneously, and to match a subset of cell clusters across the source data and the target data, we propose to solve the following optimization problem in *couple*CoC+:

$$\underset{\substack{C_Y, C_X, C_Z, C_U \\ h_{T,N_{sub}}, h_{S,N_{sub}}}}{\text{argmin}} \quad \ell_{T}(C_Y, C_Z) + \lambda \ell_{S}(C_X, C_Z) + \beta \ell_{U}(C_Y, C_U)$$

$$+ \quad \gamma D_{KL}(\hat{p}_{T}(\tilde{Y}_{h_{T,N_{sub}}}, \tilde{Z}_{T}) \| \hat{p}_{S}(\tilde{X}_{h_{S,N_{sub}}}, \tilde{Z}_{S})). \tag{5}$$

As mentioned before, the two terms $\ell_{T}(C_Y, C_Z)$ and $\ell_{S}(C_X, C_X)$ in formula (5) share the same feature cluster $C_Z$, which can be viewed as a bridge to transfer knowledge between the source data and the target data [42, 47]. The dimension of the feature space shared by the source data S and the data T is reduced by clustering and aggregating similar features. Aggregating similar features guided by the source data S enables knowledge transfer between the source data S and the data T, which reduces the noise in the single-cell data and can generally improve the clustering performance of cells in the target data. The term $\ell_{U}(C_Y, C_U)$ corresponds to the features in the target data that are unlinked with the source data. Incorporating more information from the target data by including more features will also benefit the clustering performance of the target data. The term $D_{KL}(\hat{p}_{T}(\tilde{Y}_{h_{T,N_{sub}}}, \tilde{Z}_{T}) \| \hat{p}_{S}(\tilde{X}_{h_{S,N_{sub}}}, \tilde{Z}_{S}))$ further borrows information from the source data for the matched cell clusters. $\lambda$ is a hyperparameter that controls the contribution of the source data S, $\beta$ is a hyperparameter that controls the contribution of the unlinked features in the target data, and $\gamma$ is a hyperparameter that controls the contribution of cell types matching across the source data S and the data T.

The optimization problem (5) can be solved by iteratively updating $C_X, C_Y, C_Z, C_U, h_{T,N_{sub}}$ and $h_{S,N_{sub}}$. The technical details of the updates are presented in Text A and B in S1 Text. The objective function in the optimization problem (5) is non-increasing in the updates of $C_Y, C_X, C_Z, C_U, h_{T,N_{sub}}$ and $h_{S,N_{sub}}$, and the algorithm will converge to a local minimum. Finding the global optimal solution is NP-hard. The algorithm converges in a finite number of iterations due to the finite search space (see details in Section: *Convergence and running time*). In practice, this algorithm works well in real single-cell genomic data analysis.

Lastly, we note that the major differences between *couple*CoC [42] and *couple*CoC+ lie in two aspects: (a). *couple*CoC does not include the unlinked features in the target data. We will demonstrate through the real data examples that incorporating more information by including the unlinked features will benefit clustering of the target data. (b). In *couple*CoC, cell type matching is a separate step from co-clustering. In *couple*CoC+, we simultaneously perform cell type matching and co-clustering. Merging cell type matching and co-clustering in one step can leverage more information from the source data.

## Choosing source data and target data

In *couple*CoC+, the dataset that is less sparse and less noisy should be chosen as the source data, and dataset that is more sparse and noisier should be chosen as the target data. By doing so, we expect to borrow more useful information from the source data to clustering the target data. Based on this rule, we generally choose scRNA-seq data as the source data and choose scATAC-seq data or sc-methylation data as the target data. In practice, we utilize the proportion of zero entries in the data matrix to evaluate the sparsity, and it is calculated as:

$$\frac{\# \text{ of zero entries}}{\# \text{ of cells} \times \# \text{ of features}}.$$

We will describe the details on how to choose the source data and the target data case-by-case in the real data examples.

## Feature selection

Features that are directly related to the genes are used as linked features across datasets: we use gene activity score (prefered) or promoter accessibility in scATAC-seq data, homologs in human and mouse scRNA-seq data, and gene body methylation in sc-methylation data. Features that are not directly linked to genes are treated as the unlinked features in target data: we use accessibility of peak values in scATAC-seq data and DNA methylation levels at non-CG sites for non-overlapping 100kb bins in sc-methylation data. We use the mouse specific genes that are not included in human as the unlinked features when we use mouse scRNA-seq data as the target data and human scRNA-seq data as the source data. We implement feature selection before performing clustering. We use the R toolkit Seurat [33, 34] to select 1000 most variable features for each data S, T and U.

## Data preprocessing

We take log transformation for scRNA-seq data to alleviate the effect of extreme values in the data matrices: we use log2(TPM+1) for TPM data and log2(UMI + 1) for UMI data as the input. We use gene activity score or promoter accessibility and binarized count for peaks as the input for scATAC-seq data. We use DNA methylation levels at non-CG sites in the gene body and non-overlapping 100kb bins as the input for sc-methylation data. We impute the missing values in sc-methylation data with the overall mean. Since the relationship between gene body methylation and gene expression is negative, we further transform sc-methylation data by 1-methylation level. Our proposed *couple*CoC+ can automatically adjust for sequencing depth, so we do not need to normalize for sequencing depth. The input formats of data S, T and U are described case-by-case in real data examples.

## Hyperparameter selection

Before implementing the *couple*CoC+ algorithm, we use the Calinski-Harabasz (CH) index [48] to pre-determine the number of cell clusters $N_T$ for target data and the number of cell clusters $N_S$ for source data separately. CH index is proportional to the ratio of between-clusters dispersion and within-cluster dispersion:

$$f(N) = \frac{SS_B(N)}{SS_W(N)} \times \frac{n - N}{N - 1},$$

where $SS_B(N)$ is the overall between-cluster variance, and $SS_W(N)$ is the overall within-cluster variance, $N$ is the number of cell clusters, and $n$ is the total number of cells. For each cluster number $N$, we first cluster the dataset by minimizing $\ell_T$ for target data (or $\ell_S$ for source data) by CoC (i.e. information theoretic co-clustering algorithm in [45], which is equivalent to setting $\lambda = \beta = \gamma = 0$ in formula (5)), and then calculate $SS_B(N)$, $SS_W(N)$, and obtain $f(N)$. We choose the number of cell clusters $N$ with the highest CH index.

Our *couple*CoC+ is an unsupervised learning model, and it is hard to determine the value of non-negative hyperparameters $\lambda$, $\beta$ and $\gamma$, and the number of feature clusters $K$ in data T and $K_0$ in data U in theory. In practice, we tune these parameters empirically on the datasets themselves by optimizing the CH index. Let $\Omega \triangleq (\lambda, \beta, \gamma, K, K_0)$, and we use grid search to choose the best combination of hyperparameters that has the highest CH index for the target

data: $f_T(\Omega) = \frac{SS_B(\Omega)}{SS_W(\Omega)} \times \frac{n_T - N_T}{N_T - 1}$. We choose the search domains $\lambda, \beta, \gamma \in (0, 5)$ and $K, K_0 \in (1, 20)$. Grid search performs well in real data analysis.

The value of $N_{sub}$ can be user-defined or chosen heuristically. The intuition for choosing $N_{sub}$ is that the KL divergence $D_{KL}(\hat{p}_T(\tilde{Y}_{h_{T,N_{sub}}}, \tilde{Z}_T) \| \hat{p}_S(\tilde{X}_{h_{S,N_{sub}}}, \tilde{Z}_S))$ will be larger if the clusters being matched are less similar when they are forced to be matched. $N_{sub}$ is chosen similarly as in [42]. More details are given in Text C in S1 Text. Though there is no theoretical guarantee, this heuristic approach for choosing $N_{sub}$ gives reasonable results in the real data examples.

### Evaluation metrics

We evaluate the clustering performance by normalized mutual information (NMI) and adjusted Rand index (ARI) [49]. Assume that $G$ is the known ground-truth labels of cells and $Q$ is the predicted clustering assignments, then NMI is calculated as:

$$\frac{I(Q; G)}{\sqrt{H(Q) * H(G)}},\tag{8}$$

where H is the entropy. NMI is normalized mutual information score and takes value between 0 and 1. Assume that $n$ is the total number of single cells, $n_{Q,i}$ is the number of cells assigned to the $i$-th cluster in $Q$, $n_{G,j}$ is the number of cells belonging to the $j$-th cell type in $G$, and $n_{i,j}$ is the number of overlapping cells between the $i$-th cluster in $Q$ and the $j$-th cell type in $G$. As a corrected-for-chance version of the Rand index, ARI is calculated as:

$$\frac{\sum_{ij}\binom{n_{i,j}}{2} - \left[\sum_i\binom{n_{Q,i}}{2}\sum_j\binom{n_{G,j}}{2}/\binom{n}{2}\right]}{\frac{1}{2}\left[\sum_i\binom{n_{Q,i}}{2} + \sum_j\binom{n_{G,j}}{2}\right] - \left[\sum_i\binom{n_{Q,i}}{2}\sum_j\binom{n_{G,j}}{2}\right]/\binom{n}{2}}.\tag{9}$$

The higher values of NMI and ARI indicate better clustering performance.

### Results

We evaluated our method *couple*CoC+ in four real data examples, including one example for clustering mouse cortex scATAC-seq data and scRNA-seq data, one example for clustering human and mouse scRNA-seq data, one example for clustering mouse cortex sc-methylation data and scRNA-seq data, and one example for clustering human blood dendritic cells scRNA-seq data generated from two experimental batches. UMAP visualizations of all these raw data are presented in S1–S4 Figs. We compared *couple*CoC+ with *couple*CoC [42], CoC [45], k-means and other commonly used clustering methods for single-cell genomic data, including SC3 [20], SIMLR [19], SAME-clustering [24] and SHARP [25] for scRNA-seq data, Cusanovich2018 [30] and *cis*Topic [29] for scATAC-seq data (we implemented louvain clustering after dimension reduction by Cusanovich2018 [30] and *cis*Topic [29], which was suggested in a recent benchmark study on scATAC-seq data [50]), BPRMeth-G [31] (Gaussian-based model proposed in [31]) for sc-methylation data, and Seurat [34], LIGER [38] and *sc*ACE [39] for the integrative clustering of source data and target data. For a fair comparison, we implemented the benchmarked methods (except coupleCoC) with both the linked and the unlinked features. We determined the number of cell clusters for *couple*CoC+ by the CH index, and we used the true number of cell clusters for the other methods, except for the methods Seurat [34] and LIGER [38], which automatically determine the number of cell clusters. We used ARI, NMI and the clustering table to evaluate the clustering results, where the cell type labels provided in their original publications were treated as the ground truth.

## Example 1: Integrative clustering for mouse cortex scATAC-seq data and scRNA-seq data

We first evaluated *couple*CoC+ by jointly clustering mouse cortex scATAC-seq data [30] and scRNA-seq data [51]. We collected 458 oligodendrocytes, 551 astrocytes, 319 inhibitory neurons, 197 microglia cells for the scATAC-seq data and collected six subtypes of inhibitory neurons (including 1122 Lamp5 cells, 1741 Sst cells, 1337 Pvalb cells, 125 Sncg cells, 27 Serpinfi cells and 1728 Vip cells), 368 astrocytes and 91 oligodendrocytes for the scRNA-seq data. Note that microglia cells are not used in the scRNA-seq data. We chose the scATAC-seq data as the target data, and scRNA-seq data as the source data, because scATAC-seq data is noisier and sparser (The proportions of zero entries in the scATAC-seq data and scRNA-seq data are 95.71% and 86.68%, respectively.) We used gene activity score in scATAC-seq data as the features that are linked with scRNA-seq data, and used the accessibility of the peaks as the unlinked features. The input formats are log(TPM+1) for scRNA-seq data, and binarized gene activity score and binarized peak accessibility for scATAC-seq, respectively. We used the provided cell type labels as a benchmark for evaluating the performance of the clustering methods. The numbers of cell clusters with the highest CH indexes are $N_T$ = 5 for the target data and $N_S$ = 6 for the source data (S5 Fig). In the source data, there are six subtypes of inhibitory neurons and two other cell types, and the smaller cell cluster number ($N_S$ = 6) chosen by CH index likely represents the similarity of the six subtypes of inhibitory neurons. We implemented *couple*CoC+ with $N_S$ = 8 and $N_T$ = 5. We set the tuning parameters in *couple*CoC+ as $\lambda$ = 2.5, $\beta$ = 0.01, $\gamma$ = 1, $K$ = 12, $K_0$ = 6 by grid search. We set the number of $N_{sub}$ as 4, because the objective function $g(N_{sub})$ for choosing $N_{sub}$ (The formula of $g(N_{sub})$ is given in Text C in S1 Text) obtains the minimum 0.021 when $N_{sub}$ = 4 (S6 Fig).

Table 1 shows that *couple*CoC+ performs better than *couple*CoC on clustering the target data, because *couple*CoC+ utilizes information from clustering the data U which is not present in *couple*CoC, and it performs much better than CoC, because *couple*CoC+ transfers knowledge from clustering the source data S. The methods *cis*Topic and Cusanovich2018 perform well but not as good as *couple*CoC+. The performance of clustering the source data by *couple*CoC+ is better than *couple*CoC, and ranks the third among ten clustering methods. The integrative clustering methods Seurat, LIGER and *sc*ACE perform worse than *couple*CoC+, except for clustering the source data by LIGER. The clustering table (Table A in S1 Table) by *couple*CoC+ shows that the cell types astrocytes and oligodendrocytes are matched well across the two data types. Fig 2 shows the heatmap after clustering by *couple*CoC+. *couple*CoC+ clearly clusters similar cells and features. In addition, we can see that the pairs of matched cell clusters m1–4 in the two datasets clearly resemble each other more, compared with the other unmatched cell clusters.

Next we investigated the features that are clustered together by *couple*CoC+. Feature cluster "clu4" is specific to cell cluster "clu m3" in scRNA-seq and scATAC-seq data, which are mostly oligodendrocyte cells; and feature cluster "clu6" is specific to "clu t5" in scATAC-seq data, which are mostly microglia cells. We performed functional annotation enrichment analysis using DAVID [52, 53]. The genes in feature cluster "clu4" (59 genes in total) are highly enriched for the terms related to myelin (more comprehensive list in Table B in S1 Table). The top three terms and their Bonferroni corrected *p*-values are ("myelin sheath", $1.44 \times 10^{-11}$), ("myelination", $1.12 \times 10^{-7}$) and ("structural constituent of myelin sheath", $2.51 \times 10^{-5}$), respectively. By creating myelin sheath, oligodendrocytes provide support and insulation to axons in the central nervous system. The genes in feature cluster "clu6" (198 genes in total) are highly enriched for the terms related to the immune system (more comprehensive list in Table C in S1 Table). The top two terms and their Bonferroni corrected *p*-values are

**Table 1. The results of clustering the cells in source data and target data in four examples.** Note that the capital letters in the brackets represent the input data matrices for the corresponding methods: S represents source data, T and U represent the sub-matrices for the linked and unlinked features in target data, respectively. For integrative analysis methods (*couple*CoC+, *couple*CoC, Seurat, LIGER, scACE) that utilize both the source data and the target data as input, they produce clustering results of the cells in source data and target data simultaneously. We then summarize the clustering results by calculating ARI and NMI for source data and target data separately. For the remaining methods that are implemented on only one dataset, they produce clustering results of the cells in source data or target data independently. We then summarize the clustering results by calculating ARI and NMI for source data and target data separately. The source data type is scRNA-seq data for all four examples, while the target data types for examples 1–4 are scATAC-seq data, scRNA-seq data, sc-methylation data and scRNA-seq data, respectively. The symbol "-" means that the corresponding clustering method is not designed for that data type. We only compared the methods for integrative analysis of multiple datasets in example 4. $n_T$ and $n_S$ are the numbers of cells in the target data and the source data, correspondingly. Because we included the unlinked features when implementing CoC, $k$-means, Cusanovich2018, *cis*Topic, SC3, SIMLR and BPRMeth-G, the clustering results for these methods are better than that presented in [42].

| Clustering methods | | Example 1 $(n_T = 1525, n_S = 6539)$ | | Example 2 $(n_T = 292, n_S = 171)$ | | Example 3 $(n_T = 1102, n_S = 2383)$ | | Example 4 $(n_T = 288, n_S = 288)$ | |
|---|---|---|---|---|---|---|---|---|---|
| | | ARI | NMI | ARI | NMI | ARI | NMI | ARI | NMI |
| Target data | *couple*CoC+ (T+S+U) | 0.898 | 0.886 | 0.859 | 0.769 | 0.869 | 0.782 | 0.837 | 0.804 |
| | *couple*CoC (T+S) | 0.843 | 0.850 | 0.790 | 0.672 | 0.619 | 0.500 | 0.838 | 0.800 |
| | CoC (T+U) | 0.810 | 0.879 | 0.819 | 0.728 | 0.839 | 0.743 | - | - |
| | $k$-means (T+U) | 0.479 | 0.615 | 0.636 | 0.537 | 0.738 | 0.638 | - | - |
| | Cusanovich2018 (T+U) | 0.876 | 0.824 | - | - | - | - | - | - |
| | *cis*Topic (T+U) | 0.860 | 0.855 | - | - | - | - | - | - |
| | SC3 (T+U) | - | - | 0.839 | 0.683 | - | - | - | - |
| | SIMLR (T+U) | - | - | 0.691 | 0.578 | - | - | - | - |
| | BPRMeth-G (T+U) | - | - | - | - | 0.293 | 0.193 | - | - |
| | Seurat (T+S+U) | 0.697 | 0.725 | 0.815 | 0.672 | 0.317 | 0.336 | 0.777 | 0.728 |
| | LIGER (T+S+U) | 0.564 | 0.525 | 0.434 | 0.313 | 0.358 | 0.236 | 0.729 | 0.667 |
| | scACE (T+S+U) | 0.859 | 0.855 | 0.490 | 0.371 | 0.042 | 0.016 | 0.496 | 0.479 |
| Source data | *couple*CoC+ (T+S+U) | 0.716 | 0.754 | 0.930 | 0.883 | 0.987 | 0.970 | 0.858 | 0.819 |
| | *couple*CoC (T+S) | 0.666 | 0.731 | 0.930 | 0.883 | 0.985 | 0.965 | 0.858 | 0.819 |
| | $k$-means (S) | 0.682 | 0.729 | 0.268 | 0.190 | 0.972 | 0.939 | - | - |
| | SC3 (S) | 0.464 | 0.562 | 0.953 | 0.908 | 0.987 | 0.968 | - | - |
| | SIMLR (S) | 0.508 | 0.480 | 0.481 | 0.431 | 0.977 | 0.949 | - | - |
| | SHARP (S) | 0.692 | 0.733 | 0.884 | 0.811 | 0.968 | 0.935 | - | - |
| | SAME-clustering (S) | 0.727 | 0.736 | 0.930 | 0.862 | 0.975 | 0.945 | - | - |
| | Seurat (T+S+U) | 0.631 | 0.661 | 0.862 | 0.783 | 0.905 | 0.780 | 0.830 | 0.789 |
| | LIGER (T+S+U) | 0.889 | 0.859 | -0.029 | 0.027 | 0.842 | 0.748 | 0.800 | 0.731 |
| | scACE (T+S+U) | 0.683 | 0.674 | 0.885 | 0.800 | 0.987 | 0.969 | 0.489 | 0.493 |

("immunity", $8.27 \times 10^{-22}$) and ("immune system process", $3.00 \times 10^{-19}$), respectively. Microglia represents a specialized population of macrophages-like cells in the central nervous system (CNS) considered immune sentinels that are capable of orchestrating a potent inflammatory response [54]. In summary, the genes that are clustered together by *couple*CoC+ tend to be enriched for functional annotation terms closely related to the cell clusters in which they are active.

## Example 2: Integrative clustering for mouse and human scRNA-seq data

In the second example, we examined our *couple*CoC+ in datasets across different species, i.e. human and mouse scRNA-seq data [55]. We collected 99 clara cells, 14 ependymal cells, 179 mouse pulmonary alveolar type II in the mouse scRNA-seq dataset, and we collected 113 clara cells and 58 ependymal cells in the human scRNA-seq dataset. Note that there is one cell type in the mouse scRNA-seq data that is not present in the human scRNA-seq data. We chose the human scRNA-seq data as the source data and chose the mouse scRNA-seq dataset that is

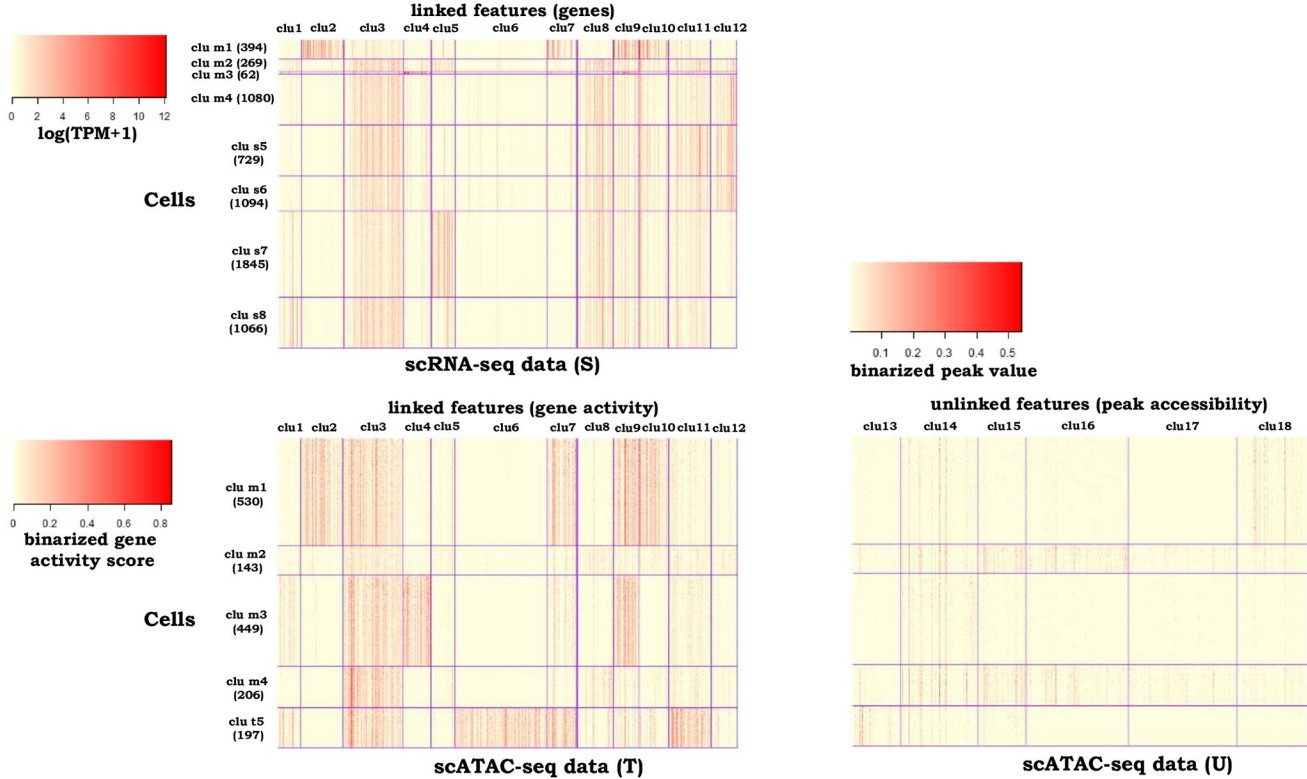

**Fig 2. Heatmaps of the clustering results by *couple*CoC+ for example 1.** "clu m" represents the matched cell cluster across the source data and the target data. "clu s" and "clu t" represent the cell clusters that are unique to the source data and the target data, respectively. For better visualization, we randomly averaged every 15 cells within the same cell cluster to generate pseudocells for every heatmap.

sparser as the target data (The proportions of zero entries in the human and mouse scRNA-seq data are 88.90% and 95.00%, respectively.). The homologs shared by mouse and human are chosen as the linked features, and mouse-specific genes are used as the unlinked features. These data are generated from the drop-seq platform, and their input formats are log(UMI+1). We use the cell type annotation [56] as a benchmark for evaluating the performance of the clustering methods. The optimal number of clusters is $N_S = 2$ for the source data, and the values of CH index are close when $N_T = 2$ or 3 (S7 Fig). We chose $N_T = 3$, which equals to the true number of cell types. We set the tuning parameters in *couple*CoC+ as $\lambda = 2$, $\beta = 0.04$, $\gamma = 1$, $K = 8$, $K_0 = 7$ by grid search. We set the number of $N_{sub}$ as 2, because the values of the objective function $g(N_{sub})$ for choosing $N_{sub}$ are smaller when $N_{sub} = 2$ (0.150 when $N_{sub} = 1$ and 0.077 when $N_{sub} = 2$, respectively).

*couple*CoC+ performs the best among all the other methods for clustering the target data (Table 1). It improves the performance over CoC by transferring the knowledge from the source data S, and also improves performance over *couple*CoC by utilizing the information in the unlinked features. SC3 has the best performance on clustering the source data, and *couple*CoC+ ranks the second. Compared to *couple*CoC+, the integrative clustering methods Seurat, LIGER and *sc*ACE do not perform well on both source data and target data. Fig 3 shows the heatmap after clustering by *couple*CoC+. *couple*CoC+ clearly clusters similar cells and features. In addition, the patterns of the linked features for the matched clusters tend to be consistent.

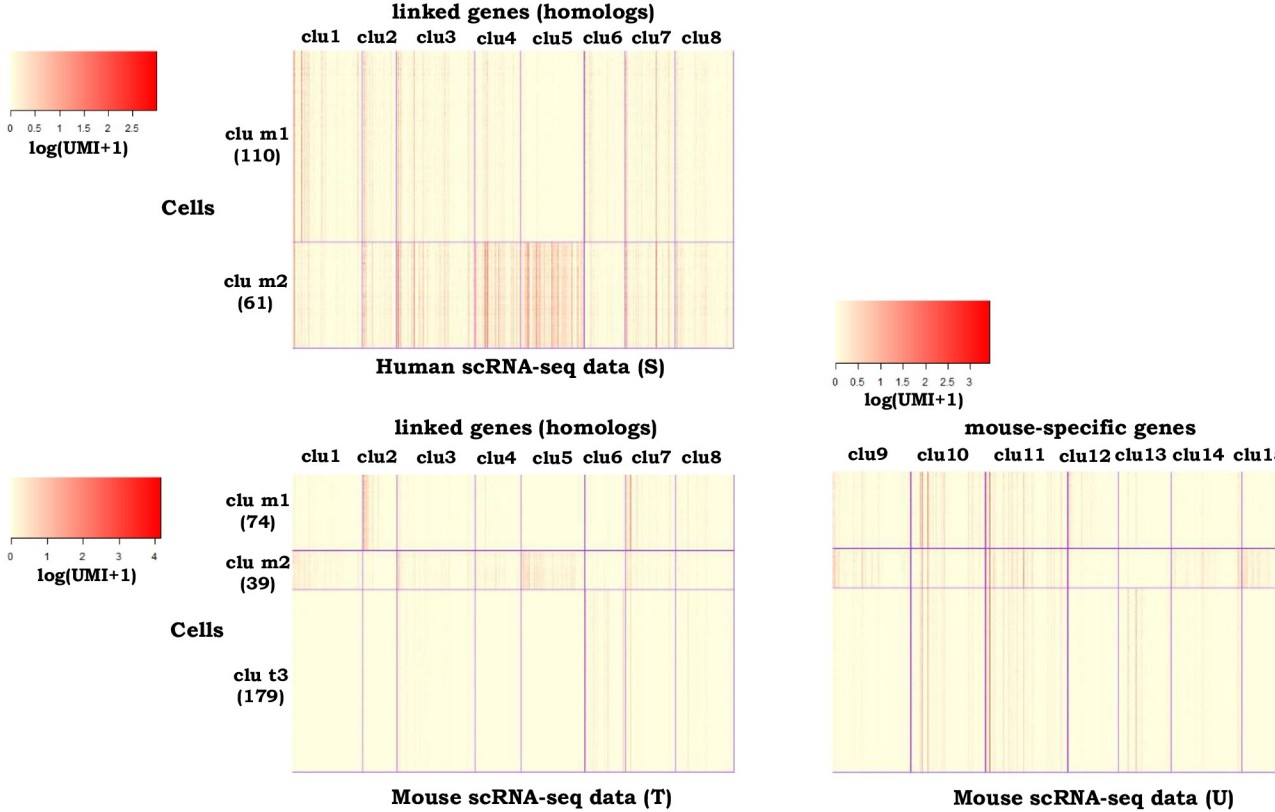

**Fig 3. Heatmaps of the clustering results by *couple*CoC+ for example 2.** "clu m" represents the matched cell cluster across the source data and the target data. "clu t" represents the cell cluster that is unique to the target data. For better visualization, we randomly averaged every 15 cells within the same cell cluster to generate pseudocells for every heatmap.

## Example 3: Integrative clustering for mouse cortex sc-methylation and scRNA-seq data

In the third example, we evaluated *couple*CoC+ by jointly clustering sc-methylation data and scRNA-seq data from the mouse cortex [7, 51]. We collected 412 L4 and 690 L2/3 sc-methylation cells, and 1401 L4 and 982 L2/3 IT scRNA-seq cells ("L4" and "L2/3" stand for excitatory neurons in different neocortical layers; IT is the abbreviation of intratelencephalic neuron.). Sc-methylation data tends to be noisier than scRNA-seq data, so we chose sc-methylation as the target data and chose scRNA-seq data as source data. The methylation of gene bodies are the linked features, and the DNA methylation levels at non-CG sites (mCH levels) for non-overlapping 100kb bins are the unlinked features. We used the provided cell type labels as a benchmark for evaluating the performance of the clustering methods. S7 Fig shows that the optimal number of cell clusters are $N_T = 2$ and $N_S = 2$. We set the tuning parameters in *couple*CoC+ as $\lambda = 0.1$, $\beta = 0.6$, $\gamma = 1$, $K = 5$, $K_0 = 8$ by grid search. We set the number of matched clusters $N_{sub}$ as 2, because the values of the objective function $g(N_{sub})$ for choosing $N_{sub}$ are smaller when $N_{sub} = 2$ (0.138 when $N_{sub} = 1$ and 0.061 when $N_{sub} = 2$, respectively).

Table 1 shows that all ten methods have good clustering performance for scRNA-seq data. *couple*CoC+ performs much better than the other methods for clustering sc-methylation data, and it matches well the cell types across the two data types (Table A in S1 Table). *couple*CoC + has better clustering performance than CoC, due to the transfer of knowledge from scRNA-seq data to clustering sc-methylation data, and *couple*CoC+ performs better than *couple*CoC,

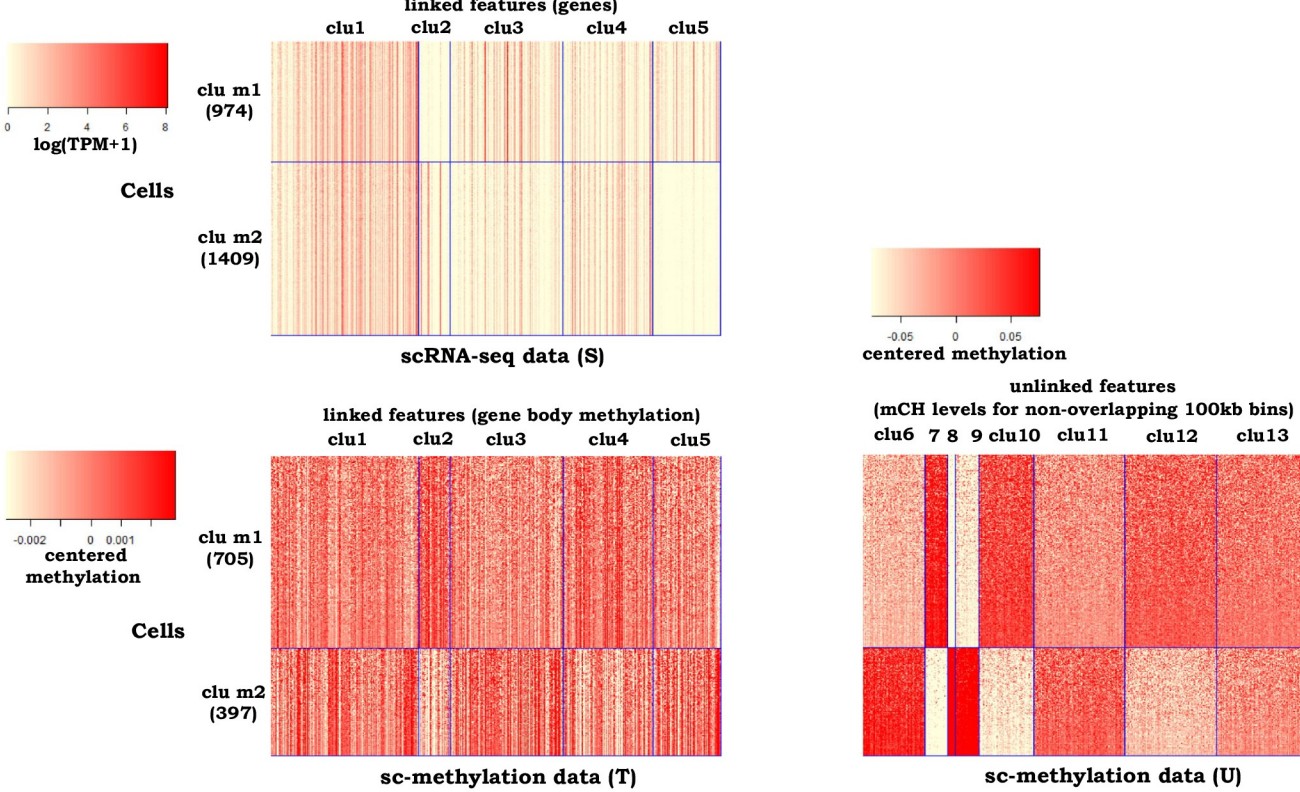

**Fig 4. Heatmaps of the clustering results by *couple*CoC+ for example 3.** "clu m" represents the matched cell cluster across the source data and the target data. We obtained the centered methylation level by first centering the data matrix by row and then centering the data matrix by column. Grey color in the heatmap of sc-methylation data corresponds to missing data. For better visualization, we randomly averaged every 15 cells within the same cell cluster to generate pseudocells for every heatmap.

because it utilizes the information in the unlinked features while *couple*CoC does not. The integrative methods Seurat, LIGER and *sc*ACE do not perform well on target data. Fig 4 is the corresponding heatmap after clustering by *couple*CoC+. For scRNA-seq data, *couple*CoC+ clearly clusters similar cells and features. The signal in sc-methylation data is weaker but we can still see the reverse trend compared with scRNA-seq data: when the gene body methylation level is lower in sc-methylation data, gene expression tends to be higher in scRNA-seq data. The heatmap of the data U further demonstrates the usefulness of including the unlinked features in sc-methylation data, where mCH levels for non-overlapping 100kb bins better distinguishes the cell types compared with gene body methylation.

## Example 4: Integrative clustering for human blood dendritic cells scRNA-seq data from two batches

In the fourth example, we examined *couple*CoC+ by integrative clustering of human blood dendritic cell (DC) scRNA-seq data from two batches [57]. Each batch consists of 96 CD141 DC, 96 CD1C DC, 96 plasmacytoid DC (pDC) and 96 double negative cells. The data were generated from the Smart-Seq2 platform and they were used in a recent benchmark study [58]. We processed the data similar to [58], where CD141 DC in batch 1 and CD1C DC in batch 2 were removed. So, both batches share pDC and double negative cells, and each batch has one unshared cell type (CD1C and CD141 respectively) that are biologically similar. We chose batch 1 as the source data and chose batch 2 that is sparser as the target data (The

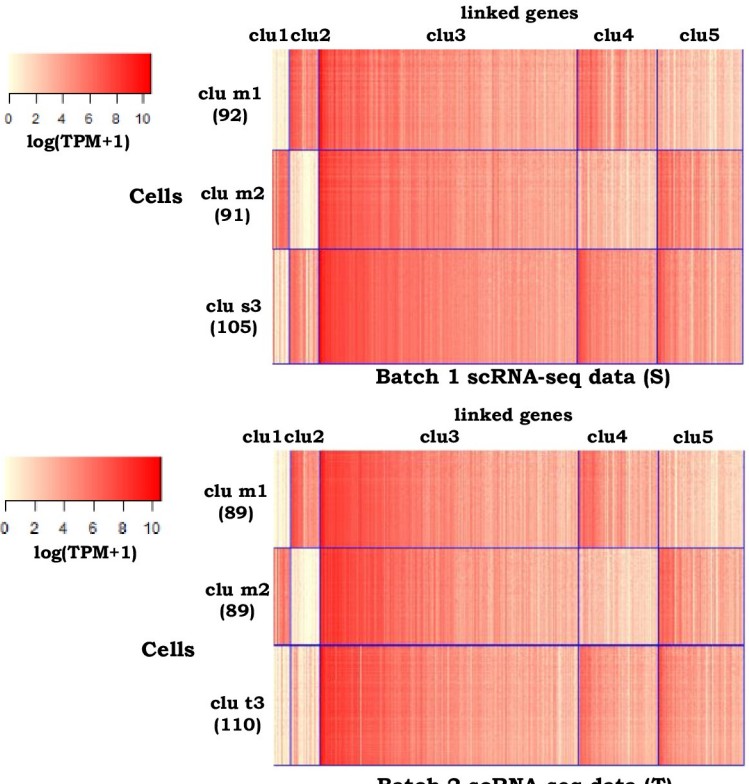

**Fig 5. Heatmaps of the clustering results by *couple*CoC+ for example 4.** "clu m" represents the matched cell cluster across the source data and the target data. "clu s" and "clu t" represent the cell clusters that are unique to the source data and the target data, respectively. For better visualization, we randomly averaged every 15 cells within the same cell cluster to generate pseudocells for every heatmap.

proportions of zero entries in batch 1 and batch 2 are 32.65% and 42.89%, respectively.). Because all features are shared by source data and target data, and target data have no unlinked features in this example, we set the value of $\beta$ as 0 in objection function (5). We set the number of cell clusters as $N_T = N_S = 3$. We set the tuning parameters in *couple*CoC+ as $\lambda = 2$, $\gamma = 1$, $K = 5$, $K_0 = 5$ by grid search. We set the number of matched clusters $N_{sub}$ as 2, because the values of the objective function $g(N_{sub})$ for choosing $N_{sub}$ are smallest when $N_{sub} = 2$: $8.66 \times 10^{-4}$ when $N_{sub} = 1$, $6.37 \times 10^{-4}$ when $N_{sub} = 2$, and $1.08 \times 10^{-3}$ when $N_{sub} = 3$.

All five integrative methods, except *sc*ACE, have good clustering performance for scRNA-seq data in batch 1 (source data)(Table 1). *sc*ACE also fails to cluster scRNA-seq data in batch 2 (target data). Because no unlinked features are included in target data, *couple*CoC+ and *couple*CoC have similar clustering performance for the data from two batches, and they have competitive performance compared with the other methods (Table 1). Fig 5 is the corresponding heatmap after clustering by *couple*CoC+. *couple*CoC+ clearly clusters similar cells and features, and it accurately found the two matched clusters (the shared pDC and double negative cells) across the two batches (Table A in S1 Table). In addition, the expression patterns for the matched clusters tend to be consistent in the two batches. The cell types unshared by the two batches, including CD1C DC and CD141 DC, are represented by "clu s3" and "clu t3", respectively. They have high similarity with each other, and they are only distinguished by feature cluster "clu2".

## Simulation studies

Lastly, we tested the performance of *couple*CoC+ through simulation studies. We followed the simulation setup given in [39] with some modifications specific to our framework. The details for generating data T, S and U are given in Text D in S1 Text. We set the numbers of cell types in both target data and source data as $N_T = N_S = 2$, set the numbers of cells as $n_T = n_S = 100$, set the proportion of each cell type as 0.5 and set the number of features as $q = q_0 = 1000$. We varied the differential degree (*w*) across clusters, the standard deviations ($\sigma_S$ and $\sigma_T$, corresponding to source data and target data, respectively) of the generative distribution, and the shift between two means (*d*) of the generative distributions for the two cell types in data U (i.e., unlinked features in target data). Larger *w* leads to better separation of the cell clusters in data S and data T, larger $\sigma_S$ or $\sigma_T$ leads to higher noise level in data S or data T, and larger *d* leads to better separation of the cell clusters in data U. We considered six different simulation settings, varying the parameters *w*, $\sigma_S$, $\sigma_T$ and *d*. We compared *couple*CoC+ with *couple*CoC and CoC.

We set the tuning parameters in *couple*CoC+ as $\lambda = 2$, $\gamma = 1$, $K = 3$, $K_0 = 3$ for all six settings, $\beta = 0.1$ for setting 5, and $\beta = 1$ for the remaining settings by grid search. We set the number of matched clusters $N_{sub}$ as 2. Table 2 presents the simulation results for the target data. In settings 1–3, we fixed $d = 2$, and we varied *w*, $\sigma_S$ and $\sigma_T$. Compared with setting 1 ($w = 0.67$, $\sigma_S = \sigma_T = 1.4$), setting 2 has higher noise ($\sigma_S = \sigma_T = 1.7$), and setting 3 has lower differential ability across the cell clusters ($w = 0.64$) in data T and data S. Compared with setting 3 ($d = 2$), the unlinked features (data U) in target data have less power in separating the cell types in setting 4 ($d = 1$). In these four settings, *couple*CoC+ performs better than *couple*CoC, because data U provide information for separating the cell types and *couple*CoC does not utilize the information in data U. In setting 4 where *d* is smaller, i.e. data U have less power in separating the cell types, the margin between *couple*CoC+ and *couple*CoC becomes smaller. Both *couple*CoC + and *couple*CoC have better clustering performance than CoC, due to the transfer of knowledge from source data to clustering target data. When data U contain no information in separating the cell types (setting 5, $d = 0$), the performance of *couple*CoC+ is slightly worse than *couple*CoC. CoC does not work well in setting 5 because it is affected by data U and it does not transfer knowledge from the source data. When source data S have higher noise (setting 6, $\sigma_S = 2.0$), the performance of *couple*CoC+ and *couple*CoC drops and they become inferior to CoC. *couple*CoC+ is slightly better than *couple*CoC in setting 6, because it incorporates the information in data U.

## Convergence and running time

*couple*CoC+ is guaranteed to converge as the objective functions in Equations (S.10-S.14) in Text A in S1 Text are non-increasing in each iteration. *couple*CoC+ tends to converge in 15

**Table 2. The simulation results for clustering the target data in 30 independent runs are summarized.** Note that the capital letters in the brackets represent the input data matrices for the corresponding methods: S represents source data, T and U represent the sub-matrices for the linked and unlinked features in target data, respectively. *couple*CoC+ and *couple*CoC utilize both the source data and the target data as input, and they produce clustering results of the cells in source data and target data simultaneously; CoC is implemented on only target data, and it produces clustering results of the cells in target data. We then summarize the clustering results by calculating ARI and NMI for target data.

| Clustering methods | Setting 1 | | Setting 2 | | Setting 3 | | Setting 4 | | Setting 5 | | Setting 6 | |
|---|---|---|---|---|---|---|---|---|---|---|---|---|
| | $w = 0.67$, $d = 2$ | | $w = 0.67$, $d = 2$ | | $w = 0.64$, $d = 2$ | | $w = 0.64$, $d = 1$ | | $w = 0.64$, $d = 0$ | | $w = 0.64$, $d = 2$ | |
| | $\sigma_S = \sigma_T = 1.4$ | | $\sigma_S = \sigma_T = 1.7$ | | $\sigma_S = \sigma_T = 1.4$ | | $\sigma_S = \sigma_T = 1.4$ | | $\sigma_S = \sigma_T = 1.4$ | | $\sigma_S = 2.0$, $\sigma_T = 1.4$ | |
| | ARI | NMI | ARI | NMI | ARI | NMI | ARI | NMI | ARI | NMI | ARI | NMI |
| *couple*CoC+(S+T+U) | 0.935 | 0.905 | 0.855 | 0.802 | 0.834 | 0.805 | 0.816 | 0.711 | 0.780 | 0.699 | 0.637 | 0.601 |
| *couple*CoC(S+T) | 0.909 | 0.874 | 0.824 | 0.750 | 0.800 | 0.746 | 0.800 | 0.746 | 0.800 | 0.746 | 0.611 | 0.555 |
| CoC(T+U) | 0.826 | 0.781 | 0.816 | 0.774 | 0.810 | 0.766 | 0.567 | 0.522 | 0.002 | 0.009 | 0.812 | 0.765 |

iterations (S8 Fig) for the four real examples. We further summarized the computation time by the methods SC3, SIMLR and *couple*CoC+ (Table D in S1 Table) in each real example. The computation time for clustering source data with ∼6.5K cells in example 1 are SC3 = 20.52 (mins), SIMLR = 55.50 (mins). The computation time when we implement *couple*CoC+ on source data S, target data T and U (with a total of ∼8.0K cells) in example 1 is 28.20 (mins). It shows that *couple*CoC+ has comparable computational speed.

Finally, in order to study the scalability of *couple*CoC+, we also examined the examples where the number of cells is much larger. We followed the procedures of data generation described in Section: *Simulation studies* by setting $q = q_0 = 1000$, and generated datasets S, T and U with $n_S + n_T = 20K$ and $50K$ cells in total. The computation time by *couple*CoC+ when the total number of cells $n_S + n_T = 20K$, $50K$ are 163.92 (mins) and 1576.05 (mins), respectively. This demonstrates that *couple*CoC+ can be implemented on datasets with 20K cells, and it can be challenging to implement *couple*CoC+ on datasets with more than 50K cells.

## Discussion

In this research, we demonstrated that *couple*CoC+, an information-theoretic co-clustering-based unsupervised transfer learning method, is useful in the integrative analysis of single-cell genomic data. First, through clustering and aggregating similar features, *couple*CoC+ implicitly incorporates dimension reduction of the feature space, which is helpful to reduce the noise in high dimensional single-cell genomic data. We empirically demonstrated that *couple*CoC+ can alleviate the problems of high dimensionality and sparsity by presenting the clustering results on real single-cell genomic datasets. Second, compared with CoC [45] and *couple*CoC [42], *couple*CoC+ yields better clustering results for target data, because it not only transfers knowledge via clustering the features that are linked with the source data but also utilizes information from the unlinked features in target data. Incorporating more information from the target data by including the unlinked features further boosts the clustering performance of the target data. Third, *couple*CoC+ can automatically find the matched cell subpopulations across source data and target data. Fourth, feature clustering by *couple*CoC+ is biologically meaningful, where it tends to group genes that are enriched for functional annotation terms closely related to the cell clusters in which they are active. Although our method *couple*CoC+ has appealing computational speed in clustering the datasets with ∼8K cells (<30 mins to implement), it is challenging to implement *couple*CoC+ on very large datasets with more than 50k cells (>24hrs to implement). Further improvement in computational speed may be achieved by optimizing the code and developing mini-batch version of the algorithm.

## Supporting information

**S1 Text. Text A**: *couple***CoC+ algorithm**. **Text B: Summary of *couple*CoC+ algorithm**. **Text C: Selecting $N_{sub}$**. **Text D: Data generation in simulation**.
(PDF)

**S1 Table. Table A**. **Clustering table by *couple*CoC+ in real data examples 1–4**. "clu m" represents the matched cell cluster across the source data and the target data. If there is no "m" in a cell cluster label, it represents that the cluster is not matched across the two datasets, and we use "clu s" and "clu t" to represent that the cluster belongs to source data and target data, respectively. **Table B**. **Enriched functional annotation terms for gene list in the "clu 4" of linked genes in example 1 using DAVID tools**. The top 10 terms are shown here. **Table C**. **Enriched functional annotation terms for gene list in the "clu 6" of linked genes in example 1 using DAVID tools**. The top 10 terms are shown here. **Table D**. **Summary of the**

**computation time by classical clustering methods SC3 and SIMLR for scRNA-seq data in examples 1–3 and by** *couple* **CoC+ for the combination of source data and target data in examples 1–4**. The algorithm *couple*CoC+ runs until convergence (15 iterations) by MATLAB R2019b—academic use. SC3 and SIMLR run in default iterations in Rstudio (Version 1.2.5033) by the downloaded R packages. All of these algorithms are run in Windows 10 Enterprise (Version 1909) with the Processor: Intel(R) Core(TM)i7–9700 CPU 3.00GHz and with 16.0 GB installed RAM.
(PDF)

**S1 Fig. UMAP visualization of source data (left) and target data (right) in example 1.**
(TIF)

**S2 Fig. UMAP visualization of source data (left) and target data (right) in example 2.**
(TIF)

**S3 Fig. UMAP visualization of source data (left) and target data (right) in example 3.**
(TIF)

**S4 Fig. UMAP visualization of source data (left) and target data (right) in example 4.**
(TIF)

**S5 Fig. Calinski-Harabasz evaluation on selecting the optimal number of cell clusters for the source dataset and the target dataset in example 1.** The value of CH index has been standardized via minimax normalization to ensure each value being bound to between 0 and 1.
(TIF)

**S6 Fig. Choose the number of $N_{sub}$ in example 1.**
(TIF)

**S7 Fig. Calinski-Harabasz evaluation on selecting the optimal number of cell clusters for the source dataset and the target dataset in examples 2 and 3.** The value of CH index has been standardized via minimax normalization to ensure each value being bound to between 0 and 1.
(TIF)

**S8 Fig. The loss function (objective function) curves after each iteration by** *couple***CoC+ in real data examples 1–4.** The value of the objective function after each iteration has been standardized via minimax normalization to ensure each value being bound to between 0 and 1.
(TIF)

## Acknowledgments

We would like to thank Jiaxuan Wangwu for her work on the implementation of integrative clustering by LIGER method.

## Author Contributions

**Conceptualization:** Pengcheng Zeng, Zhixiang Lin.

**Funding acquisition:** Zhixiang Lin.

**Methodology:** Pengcheng Zeng, Zhixiang Lin.

**Project administration:** Zhixiang Lin.

**Software:** Pengcheng Zeng.

**Supervision:** Zhixiang Lin.

**Validation:** Pengcheng Zeng.

**Visualization:** Pengcheng Zeng.

**Writing – original draft:** Pengcheng Zeng, Zhixiang Lin.

**Writing – review & editing:** Pengcheng Zeng, Zhixiang Lin.

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
