## [Decision Letter · Decision Letter 0]

23 Mar 2021

Dear Dr. Zeng,

Thank you very much for submitting your manuscript "coupleCoC+: an information-theoretic co-clustering-based transfer learning framework for the integrative analysis of single-cell genomic data" for consideration at PLOS Computational Biology.

As with all papers reviewed by the journal, your manuscript was reviewed by members of the editorial board and by several independent reviewers. In light of the reviews (below this email), we would like to invite the resubmission of a significantly-revised version that takes into account the reviewers' comments.

We cannot make any decision about publication until we have seen the revised manuscript and your response to the reviewers' comments. Your revised manuscript is also likely to be sent to reviewers for further evaluation.

Sincerely,

Qing Nie

Associate Editor

PLOS Computational Biology

Sushmita Roy

Deputy Editor

PLOS Computational Biology

Reviewer's Responses to Questions

**Comments to the Authors:**

Reviewer #1: Zeng and Lin developed coupleCoC+, a method for integrative analysis of single-cell genomic data. My comments are below.

1. The package only contains matlab scripts with minimal descriptions. Vignettes with toy examples are needed for this to be suited for publication in PLOS Computational Biology.

2. For better reproducibility, the scripts used for other methods in the benchmark also need to be uploaded. Also, some visualizations are needed for the three empirical datasets, in addition to the ARI and NMI metrics. Also, how are the metrics calculated, using what ground truths?

3. If one cluster is better separated by one modality, but not the other. How is this handled?

4. For the major figures (the heatmaps in Figure 2-4), the horizontal labels and the vertical labels overlap and can be confusing. My understanding is the horizontal labels are clustering of features, while the vertical labels are for cells – if there is no matched cell cluster in the target data, the labels overlap.

5. For integrating scRNA-seq with scATAC-seq, the authors use the “gene activity” calculated from the ATAC reads as the link features. It has been shown that the accessibility in promoter region is not always a good proxy for pseudo-gene expression. Would this be an issue? Any comments on this?

Reviewer #2: In this paper, authors proposed propose coupleCoC+, which is based on the information-theoretic co-clustering transfer learning framework for the integrative analysis of single-cell genomic data. The goal of coupleCoC+ is to utilize one dataset, the source data (S), to facilitate the analysis of another dataset, the target data. Depending on whether the features are linked with the source data or not, the target data can be partitioned into two parts, data T with the linked features, and data U with the unlinked features. The authors introduced a well-studied concept in information to solve an important computational biology problem. I have following questions.

Major comments:

1. In this article, authors use the real examples to demonstrate that the proposed method has better performance in clustering target data with source data at single-cell level. However, the simulation studies are missing to show that under which cases, coupleCoC+ is better and under which cases, coupleCoC+ does not provide gains.

2. In Table 1, it is not clear whether the summarized results are calculated by applying ARI and NMI to target or source data, or both.

3. Authors demonstrate coupleCoC+ using three real examples. I am wondering any biological interpretations to the clustering results. It would be helpful to show the biology of findings.

4. As batch effects are major obstacles to perform integrative analysis using single-cell data, I am wondering if coupleCoC+ is capable of integrating datasets from different batches. It would be helpful to discuss it.

**Have all data underlying the figures and results presented in the manuscript been provided?**

Reviewer #1: **No: **Source data and analytical scripts are not available.

Reviewer #2: Yes

PLOS authors have the option to publish the peer review history of their article (what does this mean?). If published, this will include your full peer review and any attached files.

Reviewer #1: No

Reviewer #2: No
---

## [Decision Letter · Decision Letter 1]

11 May 2021

Dear Dr. Zeng,

We are pleased to inform you that your manuscript 'coupleCoC+: an information-theoretic co-clustering-based transfer learning framework for the integrative analysis of single-cell genomic data' has been provisionally accepted for publication in PLOS Computational Biology.

Best regards,

Qing Nie

Associate Editor

PLOS Computational Biology

Sushmita Roy

Deputy Editor

PLOS Computational Biology

Reviewer's Responses to Questions

**Comments to the Authors:**

Reviewer #1: The authors have addressed all of my previous concerns.

Reviewer #2: Authors address all my concerns. I don't have additional questions.

**Have the authors made all data and (if applicable) computational code underlying the findings in their manuscript fully available?**

Reviewer #1: None

Reviewer #2: Yes

PLOS authors have the option to publish the peer review history of their article (what does this mean?). If published, this will include your full peer review and any attached files.

Reviewer #1: No

Reviewer #2: No

---

## [Editor Report · Acceptance letter]

28 May 2021

PCOMPBIOL-D-21-00237R1 

coupleCoC+: an information-theoretic co-clustering-based transfer learning framework for the integrative analysis of single-cell genomic data

Dear Dr Lin,

I am pleased to inform you that your manuscript has been formally accepted for publication in PLOS Computational Biology. Your manuscript is now with our production department and you will be notified of the publication date in due course.

With kind regards,

Zsofi Zombor
